# Factors That Foster Therapeutic Alliance in Pediatric Sports and Orthopedics: A Systematic Review

**DOI:** 10.3390/ijerph191811813

**Published:** 2022-09-19

**Authors:** Rachel N. Meyers, Robyn B. McHugh, Alissa M. Conde

**Affiliations:** 1Division of Occupational and Physical Therapy, Cincinnati Children’s Hospital Medical Center, 3333 Burnet Avenue, Cincinnati, OH 45229, USA; 2Division of Sports Medicine, Cincinnati Children’s Hospital Medical Center, 3333 Burnet Avenue, Cincinnati, OH 45229, USA

**Keywords:** therapeutic alliance, working alliance, musculoskeletal injury, healthcare provider, pediatric sports, pediatric orthopedics

## Abstract

Therapeutic alliance has been defined as building rapport between provider and patient in order to enhance patient motivation to improve outcomes. The purpose of this systematic review was to identify factors that patients look for that help build a strong therapeutic alliance in their pediatric sports or orthopedics healthcare provider, to identify if these factors differ across healthcare professions, and to identify any differences in therapeutic alliance between patients and their provider regarding in-person and telehealth visits. Scientific databases were searched from inception until August 2022. The search strategy resulted in 2195 articles with 11 studies included in the final analysis. The main attributes adolescents look for in their pediatric sports healthcare provider were shared decision making and understanding patients’ sports and goals. These factors were found to differ among parents, sex, race, and socioeconomic status. The top factors improving therapeutic alliance in telehealth were having an already established relationship with the provider, visits lasting longer than 30 min, and having an English-speaking provider for English-speaking patients. The available literature highlights factors that contribute to the development of a stronger therapeutic alliance in the pediatric sports and orthopedics population. As these factors differ among adolescents, parents, sex, race, and socioeconomic status, this review provides insight in what patients and families look for in their provider when seeking care.

## 1. Introduction

Musculoskeletal injuries have been a rising concern for the pediatric and adolescent population over recent decades [1]. In the United States, an estimated 45 million youths participate in sports programs [2], with the prevalence of injury on the rise. Approximately 5 million youths under the age of 18 are seen by their primary care or sports medicine physician for an injury each year [3], with that number not including other healthcare providers. Additionally, nearly 3.5 million youths under the age of 14 are seen each year for a sports-related injury [4], and according to the Center for Disease Control (CDC), roughly 50% of all sports injuries are preventable [5]. Consequently, musculoskeletal injuries, particularly in youths, are occurring at rates that place a high demand on our healthcare system. It is not uncommon for youths to see their healthcare provider for months at a time when facing a musculoskeletal injury, often putting a large amount of trust in their healthcare provider to help them return to specific goals and activities.

Therapeutic alliance, also known as “working alliance”, has only recently gained attention within the healthcare system over the past couple decades [6,7]. Therapeutic alliance has been defined as building rapport in order to promote patient motivation and encourages a mutual agreement for goals and treatment plans [8]. Therapeutic alliance has been shown to build a relational bond between the healthcare provider and patient, improving patient outcomes and prognosis [8,9]. Multiple studies in adults [8,9,10,11] have shown that psychosocial factors, such as therapeutic alliance, help improve musculoskeletal outcomes, especially for patients with chronic pain [10]. Additionally, Charmant et al. examined the specific factors that fostered therapeutic alliance among elite, adult athletes compared to the general population seeking physical therapy treatment [12]. This study found that the factors are drastically different in elite athletes compared to the general population. Elite athletes had different treatment goals and placed a higher emphasis on developing a personal bond with their healthcare provider compared to the general population. Although a strong therapeutic alliance has been found to improve care in adults, few studies have researched the impact of therapeutic alliance on the youth and adolescent population.

The majority of studies that have examined therapeutic alliance in youths have been for the oncology, mental health, or in-patient settings. Within pediatric oncology, there were themes that emerged to support a framework for therapeutic alliance: human connection, empathy, presence, partnering, inclusivity, humor, and honesty [13]. Among youths seeking mental health and addiction treatment, a strong therapeutic alliance had an eightfold odds of a favorable treatment outcome compared to youths with a weak therapeutic alliance [14].

Therapeutic alliance has been found to improve outcomes across all ages and specialty settings [6,7,8,9,10,11,12], and can be critical following the COVID-19 pandemic where feelings of connectedness may have been lost [15]. A survey interviewing over 7000 adolescent high school students found that 37% experienced poor mental health during the COVID-19 pandemic due to a lack of feeling connected [15]. With more youths experiencing higher mental health concerns following the pandemic [15], as well as feeling isolated when experiencing a sports injury [4], therapeutic alliance was found to be essential in improving care in adolescent athletes. To the best of our knowledge, there have been no systematic reviews published to date examining themes in the literature that highlight therapeutic alliance within the pediatric sports and orthopedics population.

This systematic review specifically identifies factors that foster therapeutic alliance in the pediatric sports and orthopedics population. The purpose of this systematic review was multi-faceted with three objectives. The primary objective was to clearly identify specific attributes that patients look for that help build a strong therapeutic alliance in their pediatric sports or orthopedics healthcare provider. The second objective was to identify if these factors that foster therapeutic alliance differ across healthcare professions in this population. The third objective was to identify any differences in therapeutic alliance between patients and their pediatric sports or orthopedics healthcare provider regarding in-person and telehealth visits. We hypothesized that specific factors that foster therapeutic alliance would be identified and that these factors would not drastically differ between healthcare professions. Lastly, we hypothesized that a stronger therapeutic alliance would be found in face-to-face visits compared to telehealth. 

## 2. Materials and Methods

### 2.1. Data Sources and Searches

We followed the Preferred Reporting Items for Systematic Reviews and Meta-Analyses (PRISMA) guidelines. The search was conducted by an independent reviewer (RNM) in MEDLINE (via PubMed), EMBASE (via Elsevier), Scopus (via Elsevier), SPORTDiscus, and CINAHL. We used keywords and subject headings representing therapeutic alliance, musculoskeletal injuries, and the pediatric population. Editorials, textbook chapters, animal studies, and non-English studies were excluded. The search covered the time frame from database inception through 6 August 2022. Reproducible search strategies can be found in Appendix A. 

### 2.2. Participants

Study participants were included if they were (1) under the age of 18 years old and (2) seeking medical care from a physician, physical therapist, athletic trainer, physician assistant, or nurse for an orthopedic, sports, or musculoskeletal injury. Exclusion criteria included: (1) patients not being treated for an orthopedic, sports, or musculoskeletal injury, (2) patients ages 18 years old or older, and (3) studies that were editorials, textbook chapters, studies not published in English, conference reports, or letters to the editors.

### 2.3. Operational Definitions

Therapeutic alliance: our operational definition for therapeutic alliance was defined as a cooperative relationship between the patient and healthcare provider [16]. Therapeutic alliance is a description of the interaction between the healthcare provider and their patients in order to optimize an environment of trust, comfort, and a positive experience. By establishing a therapeutic alliance, the provider then seeks to provide patient-centered care, in which the provider is seen as a facilitator for the patient to achieve their goals, rather than an authority figure. Therapeutic alliance encourages the patient to become more active in their treatment and to engage in a collaborative, active approach to recovery [16]. 

Musculoskeletal injury: Our operational definition for musculoskeletal injury was defined as an injury that affects the bones, muscles, ligaments, nerves, tendons, as well as concussions [17]. Some examples include fractures, physeal injuries, sprains or strains, ligament injuries or concussions. We included injuries sustained from sporting activities. We excluded injuries that were caused from cancer, cerebral palsy, or any pediatric neuromuscular disorder. 

### 2.4. Study Selection

Results from our search strategy were uploaded into Covidence (Veritas Health Innovation; www.covidence.org (accessed on 18 December 2021)), a systematic review software. Duplicate studies were immediately identified by the software and removed. Two independent reviewers (RNM, RBM) screened title and abstracts. A third independent reviewer (AMC) settled any disagreements between the two primary reviewers. 

### 2.5. Data Extraction

Two independent reviewers (RNM, RBM) extracted the data from our included studies. Data extracted included study design, patient or parent perspective, type of healthcare professional, and factors that build a strong therapeutic alliance. 

### 2.6. Quality Assessment

Risk of bias was assessed using the Downs and Black checklist for non-case reports [18]. This 26-item checklist uses four subscales with the categories of reporting, internal validity (bias), internal validity (confounding selection bias), and external validity. Two independent reviewers (RNM, RBM) completed this checklist. A third independent reviewer (AMC) settled any disagreements between the two primary reviewers. 

## 3. Results

### 3.1. Study Selection

The search strategy (Appendix A) resulted in 2195 articles. After duplicates were removed, there were a total of 1923 articles. After title and abstract screening, 48 articles were included for full text screening. Once full text screening was completed, 10 articles were included for data extraction. One additional article [19] was found with a hand-search resulting in a total of 11 articles included in the final analysis (Figure 1). 

### 3.2. Study Characteristics

Seven of the eleven included studies are cross-sectional study designs [19,20,21,22,23,24,25], one study is a qualitative study [26], one is a case report [27], one study is a randomized prospective design [28], and one is a prospective cohort design [29]. Table 1 and Table 2 includes data extracted from these included studies. Five studies included both the adolescent and parent perspectives, five studies included only the parent perspective, and one study included the adolescent perspective only. Two of the articles focused on factors associated with physical therapists, two were focused on the orthopedic care team as a whole, five focused on factors associated with physicians, and two focused on factors associated specifically with surgeons. The average age for the cross-sectional studies was 14.8 years old and the average age of patients in the qualitative study was 17 years old. The age of the athlete in the case report was 14 years old and the ages in the randomized prospective pilot study ranged from 13 to 17 years old. The average age of the prospective cohort study was 14.6 years old.

### 3.3. Factors Associated with Therapeutic Alliance

#### 3.3.1. Patient Perspective

Many factors were associated with a positive therapeutic alliance between healthcare provider and adolescents (Table 1, Figure 2). There were no differences reported among younger adolescents (10–14 years old) compared to older adolescents (15–18 years old), so results are representative of adolescent patients as a whole. The top two most important attributes to adolescents were shared decision making and understanding patients’ sports and goals [20,22,24,27,29]. Additionally, the highest valued physician characteristic from adolescent patients in a study by Beck et al., 2021 [29], was being a good listener and the lowest reported was “physician does not involve them in decision making”. Overall, 92% of all adolescent patients reported wanting to be involved in the decision-making process [20].

Specifically regarding attributes for physical therapists, adolescent patients undergoing anterior cruciate ligament (ACL) reconstruction surgery reported they want their physical therapist to be the following: a guide, motivator, booster of confidence, and coordinator of care [26]. According to Paterno et al. [26], physical therapists were often described as a resource that filled knowledge gaps and helped patients through the recovery process on both the medical and rehabilitative sides. Patients also reported that they would have benefited from a more explicit way of knowing how they were doing in terms of rehabilitation milestones with the addition of a milestone chart to track progress. 

Additionally, patients reported that they preferred their physical therapist to be a source of motivation, “almost like a coach”. Many patients commented in this study how a physical therapist was a key resource who helped them persevere through the low points and stay on track to complete rehabilitation. The physical therapist’s personality and style was found to play a significant role in patient’s perception of the physical therapist as a motivator. 

The friendliness of the healthcare provider was also found to be an important attribute regarding therapeutic alliance across multiple studies [22,27]. Patients also reported the importance of their physical therapist to be a booster of confidence, particularly in their final phase of rehabilitation when returning to sport. Lastly, a final attribute that patients sought in their physical therapist was a coordinator of care, particularly when working with athletic trainers and sports medicine physicians. In this study [26], many patients described that a physical therapist served as the primary broker in the communication and coordination of care to other healthcare providers. 

**Table 1 ijerph-19-11813-t001:** Reports mentioning parent, adolescents, or both perspectives, and factors associated with therapeutic alliance.

Study	Study Design, ROB Score	Outcome Measure	Provider	Patient Perspective	Parent Perspective	Telehealth	Factors
Beck et al., 2019 [20]	Cross-sectional, **12**	Survey	Surgeon	x	x		**Shared decision making** **Understanding sports and goals Treating like an individual** **Provider recommendations** **Good listener** Provider sex Demonstrates assertiveness Demonstrates compassion
Paterno et al., 2019 [26]	Qualitative, **7**	Interview	PT	x	x		**Provider instilling confidence** **Provider being a coordinator of care** Provider being a guide Provider being a motivator
Allison et al., 2022 [21]	Cross-sectional, **12**	TSUQ, WAI	Physician	x	x	x	Having a pre-established relationship with provider
Elbin et al., 2021 [28]	Randomized prospective, **16**	TASCP, TASC-R	Physician	x	x	x	In-person visits for parents
Beck et al., 2021 [29]	Prospective cohort, **12**	Survey	Physician	x	x		**Shared decision making** **Understanding goals** **Treating like an individual** **Provider recommendations** **Good listener** Physician schooling
Peng et al., 2018 [22]	Cross-sectional, **8**	Survey	Orthopedic team		x		**Shared decision making** **Effective communicator** **Provider recommendations**
Singleton et al., 2021 [23]	Cross-sectional, **13**	CARE measure	Physician		x		Empathy
Singleton et al., 2022 [19]	Cross-sectional, **12**	CARE measure	Physician		x		**Good listener** Respect for caregiver’s thoughts
Adado et al., 2021 [24]	Cross-sectional, **10**	PCC model	Orthopedic team		x		**Shared decision making** **Understanding patient’s needs and goals** **Being an effective communicator and educator** **Being a coordinator of care** **Making the child feel safe and comfortable** Provide emotional support to the child Involving the child’s support system in the plan of care Setting up a continuing care plan after each visit
Hanna et al., 2021 [25]	Cross-sectional, **13**	PSQ, TUQ	Surgeon		x	x	Treatment time > 30 min English-speaking provider for English-speaking patients
VanEtten et al., 2021 [27]	Case report, **NA**	Stages technique	PT	x			**Shared decision making** **Congruence in goals** **Treating like an individual** **Effective communicator** **Connectedness/friendliness** Giving clear expectations of recovery Establishing roles and responsibilities

Key: Blue = both parent and patient perspective, Green = parent perspective only, Red = patient perspective only, ROB = Risk of Bias. Note: The x in this table represents if a study mentions the patient perspective, parent perspective, or telehealth. Attributes in bold are reported to be listed in more than one article as an important factor associated with therapeutic alliance.

#### 3.3.2. Male vs. Female Patient Perspective

When comparing male vs. female adolescents, a greater percentage of female patients reported that surgeon compassion (72% vs. 47%; *p* < 0.001), physician schooling (*p* < 0.003), and being a good listener (82% vs. 61%; *p* < 0.001) were very important [20]. Additionally, a greater proportion of female patients (70%) than male patients (58%) reported that being treated as an individual was very important. Female patients (47%) also reported that surgeon assertiveness was a more important factor than male patients (34%); *p* < 0.03. Although both female and male patients reported that hearing good things about the surgeon was very important, female patients (77%) ranked this attribute higher than male patients (65%); *p* = 0.045. Surgeon sex was reported as one of the least important attributes for therapeutic alliance; however, studies examining this factor found mixed results. In Beck et al., 2019 [20], more male patients (26%) reported preferences towards surgeon sex compared to female patients (12%); *p* = 0.005. Among the male patients, 25% reported they would prefer a male surgeon, less than 1% reported preference for a female surgeon, and 74% had no preference. Among the female patients, 8% reported preference for a female surgeon, 4% preferred a male surgeon, and 88% had no preference. On the other hand, in a study by Beck et al., 2021 [29], female patients (36%) and female parents (22%) were more likely to report having a preferred physician gender compared to male patients (20%, *p* < 0.001) and male parents (15.7%, *p* < 0.001). Of those who had a gender preference, female patients and parents often preferred female physicians over male physicians, whereas male patients and parents often preferred male physicians (*p* < 0.001). 

### 3.4. Parent Perspective

In Beck et al.’s 2019 study [20], only fair to moderate agreement was seen involving the importance of surgeon characteristics between parent and patient. Patients and parents both reported being involved in the decision-making process as the most important surgeon characteristic, followed by understanding patient’s sports and goals as the second most important surgeon characteristic. Unlike adolescents reporting hearing good things about the surgeon as the third most important attribute, parents reported treating their child as a unique individual as the third most important characteristic. There was fair agreement that the most important qualities included shared decision making and understanding their sports and goals (kappa 0.236). Very few parents reported surgeon sex as an important attribute to therapeutic alliance. However, it was found to be a rare occurrence that both the parent and patient would choose their surgeon or physician together (33% from Beck et al., 2019 [20] and only 6% from Beck et al., 2021 [29]). According to Beck et al., 2021 [29], almost half of parents reported that the patient’s primary care physician or insurance company determined their sports medicine physician without their input, whereas 43% reported that the parent chose the physician. When both the patient and parent chose the provider together, no statistically significant differences were found in surgeon preferences; however, when parents made the decision alone, the patients were more likely to have different surgeon preferences from their parents, specifically on the issue of surgeon sex (*p* < 0.05). 

In Singleton et al.’s 2021 study [23], only perceived physician empathy (as measured by the Consultation and Relational Empathy Measure score) was significantly correlated with parent satisfaction (*p* < 0.0001). However, in Singleton et al.’s 2022 study [19], the most significant determinants of perceived physician empathy in pediatric orthopedic surgery are whether the parent felt listened to during the encounter and whether the physician showed respect for the parents’ thoughts. In this study, parent demographics, health literacy, self-rated mental health, wait time, and time spent with the physician did not significantly affect perceived physician empathy. 

Additionally, in Peng et al.’s study [22], the factors most predictive of satisfaction from parents were “staff working together as a team” (r = 0.82), “friendliness of provider” (r = 0.80), “cheerfulness of practice” (r = 0.80), and “likelihood of recommending provider” (0.80). In Adado et al.’s study [24], parents reported the most important qualities were factors involved in patient-centered care which included shared decision making, respect of patient’s needs and goals, being a coordinator of care, being an effective communicator and educator, making the child feel safe and comfortable, providing emotional support to the child, involving the child’s support system in the plan of care, and setting up a continuing care plan after each visit. Overall, parents cared most about the staff being team players and providing a safe and comfortable environment for their child.

### 3.5. Race and Socioeconomic Status Perspective 

Beck et al. 2021 [29], examined patients’ preference for sports medicine physician characteristics, based on patient race and socioeconomic status. Parents and guardians who were Hispanic, non-White, non-English-speaking, who had government or no insurance (compared to private insurance), or less than college level of education reported a significantly greater importance of the physician independently determining the treatment plan (*p* < 0.001). Additionally, patients self-reporting as Asian/Other/Multiracial had statistically significant lower importance scores for the provider being a good listener, being treated as an individual, and hearing positive things about the physician (*p* < 0.05) compared to their parents. African American patients and parents reported the highest scores for being a good listener, understanding patient goals, and the patient being treated like an individual; however, these differences were not statistically significant between patients self-identifying as White, Hispanic, or African American. 

In the Beck et al. study [29], no difference in physician gender preference based on language or level of education of the parent was found. However, parents of patients with no insurance or government insurance were more likely to state a preferred physician gender (26%) than the parents of private insurance (11%, *p* < 0.004). Additionally, parents of Hispanic and African American patients were more likely to have a preferred physician gender than parents of White patients (*p* = 0.048). Hispanic parents equally preferred male and female physicians, while parents of African American patients preferred male physicians. The patients’ preferred physician gender did not differ based on race or insurance type.

### 3.6. Differences among Healthcare Providers

No studies examined the differences of therapeutic alliance factors among healthcare providers in the pediatric sports and orthopedics population. Two of the articles focused on factors associated with physical therapists [26,27], two were focused on the orthopedic care team as a whole [22,24], five focused on factors associated with physicians [19,21,23,28,29], and two focused on factors associated specifically with surgeons [20,25]. Very few differences were found among healthcare disciplines. Specifically for surgeons, provider sex was listed as the least important attribute. Specifically for physical therapists, being a guide, motivator, and coordinator of care were reported to be the most important attributes. Across all disciplines, shared decision making, understanding the patients’ goals, friendliness of provider, making the patient feel safe and comfortable, and being an effective communicator were listed as the top attributes to foster a strong therapeutic alliance between provider and patient. 

### 3.7. Therapeutic Alliance Regarding Telehealth vs. In-Person

Three included studies examined the association of therapeutic alliance in telehealth visits vs. in-person visits [21,25,28]. Therapeutic alliance scores, measured by the Therapeutic Alliance Scale for Children-Revised (TASC-R), were not statistically different for adolescent patients with concussions in the in-person or telehealth setting [28]. However, therapeutic alliance scores for parents, measured by the Therapeutic Alliance Scale for Caregivers and Parents (TASCP), were significantly higher for the in-person visits compared to the telehealth visits [28]. 

Furthermore, in Allison et al.’s study [21], a stronger therapeutic alliance was measured with a higher working alliance inventory (WAI) score. In adolescents, higher continuous WAI scores were significantly associated with the telehealth visit not being their first meeting with their provider (*p* = 0.03). Higher WAI scores were also associated with greater perceived privacy (*p* = 0.028). Furthermore, some adolescents reported that having a computer screen between the patient and the provider made communication easier, alleviated vulnerability, and reduced anxiety. Several adolescents reported that telehealth felt more streamlined or instrumental than in-person visits and some adolescents described providers in telehealth as less distracted and giving more individual attention to the patient. However, some parents noted that there was less “small talk” through telehealth, and some parents noted increased distractibility by the home environment with telehealth. The majority of adolescents reported feeling comfortable talking to their provider via telehealth with one adolescent reporting their discomfort meeting with the provider was not related to transitioning to telehealth, but because they were unfamiliar with that provider. Many adolescents emphasized the importance of the long-term relationship with their provider and that these established relationships contributed to a more positive experience with telehealth [21]. 

Hanna et al. [25] described patient satisfaction in telehealth. They found that a higher patient satisfaction score was positively correlated with the length of the telemedicine visit, particularly visits that lasted more than 30 min compared to visits that lasted less than 30 min (*p* < 0.0001). The distance between the patient’s home and clinic as well as visit diagnosis were not statistically significant. However, English-speaking providers had a higher patient satisfaction score compared to their non-English-speaking counterparts (*p* = 0.017). Providers who spoke English had higher scores concerning the length of time spent with the surgeon (*p* = 0.005) and the explanation provided by the surgeon (*p* = 0.016) compared to providers who did not speak English. Education level and prior use of telemedicine did not affect patient satisfaction scores. 

**Table 2 ijerph-19-11813-t002:** Factors positively associated with therapeutic alliance in telehealth visits.

Factors Positively Associated with Therapeutic Alliance in Telehealth		
Having an already established relationship with the provider (Allison et al., 2021 [21])	Treatment time > 30 min (Hanna et al., 2021 [25])	English-speaking provider (Hanna et al., 2021 [25])

### 3.8. Risk of Bias

The second column of Table 1 displays the results of the risk of bias assessment using the Downs and Black checklist for reporting [18]. Scores ranged from 7–16. The following scores on the Downs and Black checklist have been suggested as categories of study quality: excellent 26–28, good 20–25, fair 15–19, and poor < 14 [30]. Most of the bias in our studies were due to the lack of internal validity (specifically confounding selection bias), as the majority of our included studies were cross-sectional survey designs and not randomized control trials requiring different intervention groups.

## 4. Discussion

The purpose of this systematic review was to clearly identify specific factors that patients look for that help build a strong therapeutic alliance with their pediatric sports or orthopedics healthcare provider. The second objective was to identify if these factors that foster therapeutic alliance differ across healthcare professions in the pediatric sports and orthopedic population. The third objective was to identify any differences in therapeutic alliance between patients and their pediatric sports or orthopedics healthcare provider regarding in-person and telehealth visits. All of our studies were recent, with the majority of our studies published in the last year, and the oldest study dating to 2018. We found many factors associated with a positive therapeutic alliance between healthcare provider and adolescents. The top three most important attributes included shared decision making, the provider understanding the patients’ sports and goals, and the provider being a good communicator and listener. Our systematic review found different therapeutic alliance factors based on patient and parent perspectives, sex, race, socioeconomic status, and visit type. 

### 4.1. Patient Perspective

Out of our included studies, only VanEtten et al. [27] focused on adolescents’ perspectives and five studies [20,21,26,28,29] discussed the differences between both adolescent and parent perspectives. Although shared decision making was the top reported attribute [20,22,24,27,29] and overall, 92–95% of patients wanted to be involved in the decision-making process [20,29], physician selection was determined by the parent 65% of the time, by the patient 2% of the time, and by both parent and patient together only 6–33% of the time [20,29]. It is surprising that although the vast majority of adolescents wish to be a part of the decision-making process, very few report having a say in choosing their provider. This could be due to the fact that often times, the healthcare provider is randomly assigned to the patient based on provider availability. Other reasons could be that many adolescents may not take the time to review providers’ profiles online or may not know how to decide on a provider. Additionally, some online sites may not have profiles available discussing the characteristics and special interests for each provider. Future studies should look at differences in outcomes when actively choosing the provider vs. being randomly assigned a provider and how that affects therapeutic alliance.

Additionally, being a good listener and effective communicator was also found to be an important attribute regarding therapeutic alliance across multiple studies [19,20,22,24,27,29]. This is consistent with other literature suggesting that listening to what patients have to say, asking questions, and showing sensitivity to patients’ emotional concerns was positively correlated to a high therapeutic alliance [6,7,16,31]. Additionally, physical therapists being a booster of confidence and a motivator was especially important for adolescents undergoing long rehabilitation such as surgical procedures [26]. 

### 4.2. Male vs. Female Patient Perspective

Two included studies examined the differences between males and females regarding preferences for providers [20,29]. A greater percentage of female patients reported that surgeon compassion, surgeon assertiveness, physician schooling, hearing good things about the provider, being a good listener, and being treated as an individual was very important [20]. However, studies were conflicting on male vs. female patient preference for surgeon sex. Although both males and females reported these attributes as important, it was surprising that surgeon assertiveness was listed as an important attribute among female patients when other studies suggest a calmer demeanor and not being talked down to as more important to youths [32,33]. The most common surgical procedure among adolescent females is anterior cruciate ligament reconstruction [34,35], which involves a long rehabilitation and recovery process often lasting up to a year [36]. Many adolescent females are overwhelmed with emotions and feelings of fear and anxiety when making surgical decisions [37]. Surgeon assertiveness might help ease adolescent females’ doubts and feelings of worry when the surgeon exudes confidence and assertiveness. This finding conflicts with patients’ preferences towards physical therapists in our study. Patients reported wanting their physical therapist to be a source of motivation and encouragement, “almost like a coach” [26]. One adolescent female patient reported that during ACL recovery, “I really like having a fun and encouraging therapist…that helps me a lot” [26]. Additionally, adolescent patients discuss the physical therapist as “the person patients love talking to no matter what” [26]. However, adolescent female patients also reported the importance of their physical therapist to be a booster of confidence, particularly in the patient’s final phase of rehabilitation. One female patient stated, “I feel a little bit more confident because my physical therapist told me I would be able to get back to my sport. So, knowing without a doubt I would be able to go back made it all better” [26]. Our findings suggest that adolescents prefer their surgeon to exude assertiveness and confidence, while they prefer their physical therapists to be a booster of confidence and source of motivation, particularly through long stages of rehabilitation.

### 4.3. Parent Perspective

Unlike adolescents reporting hearing good things about the surgeon as the third most important attribute, parents reported treating their child as a unique individual as the third most important characteristic [20]. Additionally, physician empathy was positively correlated with a high therapeutic alliance from parents [19,23]. It was surprising that adolescents reported hearing good things and physician schooling as an important provider attribute; however, this was not listed among parents. We hypothesized that this attribute would have been important among parents but not adolescents. Fabien et al. suggest that teammates provide significant emotional support for athletes when injured [38]. It could be suggested that adolescents care about seeing providers that their friends or teammates are seeing because that could provide a sense of increased connectedness when already feeling isolated from sports participation.

### 4.4. Race and Socioeconomic Status Perspective

Beck et al. [29] examined sports medicine physician preferences based on race and socioeconomic statuses. Parents and guardians who were Hispanic, non-White, non-English-speaking, who had government or no insurance (compared to private insurance), or less than college level of education reported a significantly greater importance of the physician independently determining the treatment plan. One reason supporting this finding could be language barriers and cultural differences. For example, those who do not speak the language of the provider may feel more comfortable with their provider independently determining the treatment plan because of fear misinterpreting results or important information. This study did not provide details if an in-person or video interpreter was utilized, which could have also affected results. Furthermore, in other cultures, patients may prefer to have their plan of care dictated by their healthcare provider. 

Additionally, parents of patients with no insurance or government insurance were more likely to state a preferred physician gender than the parents of private insurance [29]. Parents of Hispanic and African American patients were also more likely to have a preferred physician gender than parents of White patients. On the other hand, the patients’ preferred physician sex did not differ based on race or insurance type. This is consistent with literature that found gender preferences being influenced by race/ethnicity, grade level, and risk status [39]. Kapphahn et al. [39] found that being White was associated with an increase in the proportion of boys preferring a male provider, while African American, Hispanic-black, and Hispanic-white ethnicity was associated with an increased preference for female providers. Parents of different race and ethnicity could have a preferred gender among providers because of cultural differences or an increased feeling of comfort when the provider demonstrates a similar background. More studies are needed to determine if adolescents truly do not have a gender provider preference as our one included study cannot make a definitive conclusion. 

### 4.5. Differences among Healthcare Providers

No studies examined the differences of therapeutic alliance factors among healthcare providers in the pediatric sports and orthopedics population. Although we had included studies that examined provider attributes, each study only focused on one provider type. Perhaps none of the included articles focused on differences among healthcare providers because of limited resources. For example, not all patients who seek care from a physician would also necessarily seek care from a physical therapist or athletic trainer. Additionally, not all schools and sports teams have access to an athletic trainer, particularly in smaller schools, more rural environments, or schools and teams with limited budgets. Lastly, it may be suggested that a multi-disciplinary approach is more likely to occur in a hospital setting compared to private practice settings. With a more recent multi-disciplinary approach to healthcare, it is essential to determine what therapeutic alliance factors parents and patients look for in each provider of their care team. 

### 4.6. Therapeutic Alliance Regarding Telehealth vs. In-Person

Three of our included studies examined the association of therapeutic alliance with telehealth visits compared to in-person visits [21,25,28]. Telehealth visits have increased drastically in popularity during the COVID-19 pandemic due to increased exposure to sickness, recent business shut-downs, and transportation challenges [21,25,28,40]. Our study found that telehealth visits still foster therapeutic alliance among patients and parents compared to in-person visits. Three factors were found to improve therapeutic alliance for telehealth visits, such as having an already established relationship with the provider, the visit lasting longer than 30 min, and having an English-speaking provider for English-speaking patients [21,25]. Additionally, all included telehealth studies were performed by surgeons or physicians and cannot necessarily be generalized to other healthcare disciplines such as physical therapists or athletic trainers. 

The majority of sports medicine telehealth visits today last less than 30 min [41,42]. Tenforde et al. [41] sent out surveys to 119 sports medicine patients and 14 sports medicine physicians and found that the most common telemedicine visit length lasted between 15 and 29 min. Additionally, Atanda et al. [42] examined telehealth visits particularly in the pediatric sports medicine population and found that the average telemedicine visit length was 17 min. Our included telehealth studies did not go into detail on the specific goals of the visit sessions. For example, we are unaware if the sessions were for MRI reviews, which would require more time spent with the patient, or for a follow-up or quick check in with the patient, which would inherently require less time with the patient. A study by Ramani et al. [43] found that the patient history helps lead to the final diagnosis about 75% of the time. However, Ramsey et al. [44] found that primary care physicians asked only 59% of essential history items due to not spending enough time on the subjective portion of the exam. Furthermore, in a study by Acheson et al. [45], family history was discussed during 51% of visits by new patients and 22% of visits with established patients, and the average time spent on the history-taking portion of the exam was less than 2.5 min. These findings found many physicians missing a large number of items that should have been influential in developing and diagnosing treatment plans. It is concerning that most telehealth visits in our healthcare system last less than 30 min and our findings suggest that a longer telehealth session may improve therapeutic alliance and outcomes among patient and provider. 

It is also important to note that having an already established relationship with the provider prior to telehealth was positively correlated with therapeutic alliance scores [21]. Our findings suggest that telehealth may be best for follow-up appointments only once a relationship with the provider is already established and new patient visits should be completed in-person. Lastly, having an English-speaking provider for English-speaking patients was also found to enhance therapeutic alliance with telehealth visits, likely due to the ease of hearing the natural language the patient speaks. 

### 4.7. Limitations

One limitation of this systematic review is the scarcity of studies found, with many of our articles limited to non-open ended survey questions, which inherently contain bias. Furthermore, all included studies utilized surveys or outcome measures to determine therapeutic alliance factors among patient and provider. However, none of the included studies utilized surveys that were previously validated. Only one outcome measure (Therapeutic Alliance Scale for Children-Revised (TASC-R)) utilized has been validated in the pediatric population. These measures could have affected the results in our study particularly when identifying factors that patients seek in their provider. 

Additionally, all of our included telehealth studies were only on concussion patients, limiting the ability to generalize to other populations. Perhaps for a different diagnosis, such as low back pain, where therapeutic alliance has been found to significantly help prognosis and outcomes, telehealth may not be an effective option in adolescents. The demand for telehealth is increasing among health systems, and our findings should be taken into consideration when deciding on telehealth utilization for patients.

Nevertheless, this systematic review still contains valuable insight into specific factors that patients and parents look for in their healthcare provider based on sex, race, socioeconomic status, and visit type. Additionally, this systematic review was the first to examine attributes in the literature on therapeutic alliance specifically in the pediatric sports and orthopedic population as well as the first to determine therapeutic alliance of telehealth vs. in-person visits within this population.

### 4.8. Future Directions

There are many future study ideas that arose from the findings of this systematic review. Future studies should examine if therapeutic alliance attributes and factors differ across disciplines for adolescents seeking a multi-disciplinary care team. For example, examining if the same patient seeks different provider characteristics in their sports medicine physician, physical therapist, or other allied healthcare professional. Additionally, studies should ask patients a mix of open-ended survey questions and already validated multiple choice questions in order to use a gold standard and reduce the risk of implicit bias. Furthermore, studies should look at the differences of therapeutic alliance in patients with different diagnoses (perhaps surgical patients vs. non-operative patients, diagnoses of different body regions), differences based on sport type, a repeat injury or a first-time injury, age of both patient and healthcare professional, and telehealth visits that are non-concussions. There also needs to be more studies examining therapeutic alliance based on race and ethnicity as our one included study cannot make any definitive conclusions. Lastly, our study showed implications to further look into how therapeutic alliance is affected in patients who actively choose their provider compared to patients who are randomly matched with their provider. 

## 5. Conclusions

In this systematic review, we were able to identify and examine the literature on therapeutic alliance factors in the pediatric sports and orthopedics population. The main finding of our study was that shared decision making and understanding patients’ sports and goals were reported to be the top two most important attributes to adolescents. Furthermore, therapeutic alliance factors differed among patients and parents of different sex, race, ethnicity, and socioeconomic statuses. Additionally, therapeutic alliance was found to still be high in telehealth visits compared to in-person visits. The top factors improving therapeutic alliance in telehealth were having an already established relationship with the provider, visits lasting longer than 30 min, and having an English-speaking provider for English-speaking patients. The available literature highlights the importance of therapeutic alliance in the pediatric sports and orthopedics population and identifies what attributes adolescents seek when being provided care. Healthcare professionals should utilize factors that patients seek in their provider based on age, sex, race, socioeconomic status, and mode of care in order to optimize patient and family experience and improve outcomes.

## Figures and Tables

**Figure 1 ijerph-19-11813-f001:**
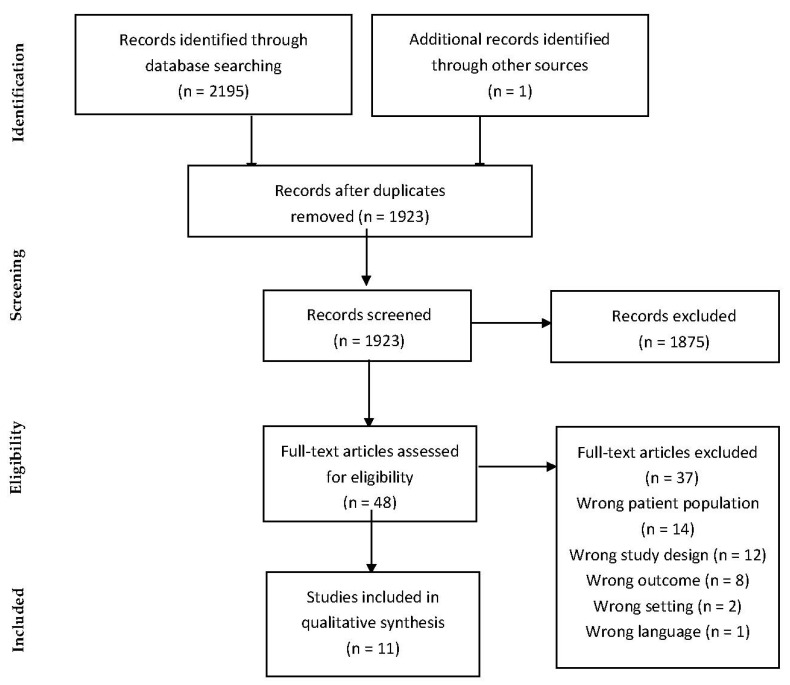
Preferred Reporting Items for Systematic Reviews and Meta-Analyses (PRISMA) flow diagram.

**Figure 2 ijerph-19-11813-f002:**
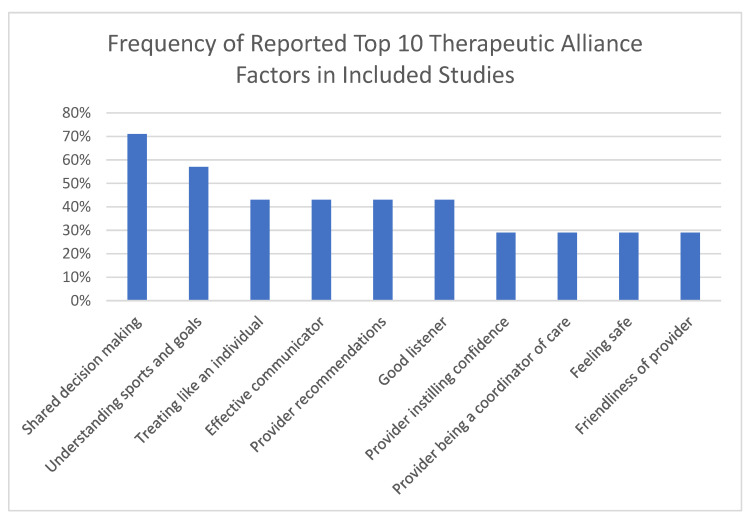
Frequency of top 10 reported therapeutic alliance factors in included studies. Note: Percentage on the y-axis represents the percent of included studies that reported the listed attributes. The x axis represents the top 10 most common listed attributes in the included studies with shared decision making being the most common.

## Data Availability

Not applicable. Search strategy provided in Appendix A.

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
