# Peer review of "Factors That Foster Therapeutic Alliance in Pediatric Sports and Orthopedics: A Systematic Review"

_ijerph, 2022, doi:10.3390/ijerph191811813_

Round 1
Reviewer 1 Report
The work is interesting and well planned.
The title reflects the content well.
The summary could be improved by showing the content of the different sections in a more individualized way (introduction, methods, results, conclusions)
The methodology is correct, clear, well stated, complete and guarantees reproducibility, as well as a validated and consistent methodology.
The results are well stated, without important biases, well founded, and in a clear and understandable way.
The limitations are very well described and analyzed.
The discussion could be briefly expanded with a comparison of similar works elsewhere.
The conclusion is clear, precise and well founded (without bias).
The figures and tables are appropriate and well designed.
Only figure 3 (Word Cloud of Most Commonly Reported Attributes) seems to me not appropriate for the manuscript. Although it may seem visually pleasing and innovative for the usual graphic styles, it should be noted that this type of image is more useful for an advertising resource than to show academic data, since the location of the words can influence the perception and value attached to each. On the other hand, although the participation of numerical variables is expressed based on the size of the typeface, it would be more precise to state it in terms of a quantifiable value. I consider this a minor correction that could enrich the manuscript.
Reviewer 2 Report
This article presents a systematic review of studies addressing the concept of therapeutic alliance in the pediatric sports and orthopedics population. The article appears to be unique in this regard. It contributes to the literature by identifying factors associated with the development of therapeutic alliance. Eleven (11) articles were included from 2196 identified after search, sufficient for this review. The article follows PRISMA guidelines. The information presented is relevant and important in health care, especially with current emphasis on patient centered care and shared decision-making. The concept of therapeutic alliance for the physical therapist is essential, as the therapist functions as a coach and guide to wellness. Future directions section reveals there is a wide range of investigations open regarding this important concept.
Specific Comments:
1. Please create a table with all data
Study, design, patient/parent perspective, type of HC prof, factors, telehealth, ROB score. If the table can be fit on a page longitudinally, it would improve a visual comparison of the articles. Combine Tables 1 +2
Columns of gender and race can be eliminated form this table, put factors in.
2. Frequency of Reported Top 10 Therapeutic Alliance Factors Figure 2: X axis labels need to be either shortened or modified to give complete information so the reader understands what is being depicted.
3. Eliminate the word cloud Figure 3. Nice for a quick visual of responses in Zoom meetings and conferences but does not give information needed for the scientific article. Better to have a graphic representation you have already included Figure 2 above which has the same information. Alternately, use pie chart.
5. Please be specific about the dates for your literature search.
6. Instead of “one study…” consistently cite the source in the body of the work, not the reference number.
7. line 359 page 11, states “…the last column of table 1 gives the results of the risk bias…”. Check Table, the scores are listed with the study design. BUT that being said…
8. Did you use Downs and Black or the Modified Downs and Black checklist (as stated)? You describe a 26 item checklist…which is NOT the Modified version.
Please explain the outcomes. Ex: Where was the bias? Was there bias in internal validity? External validity? How did you determine this what did you base this on? You start to do this in section 4.7 but can you elucidate and add to this?
9. Sometimes your wording is a little unclear. For example, page 9 line 288 is confusing, I think it should read, “…examined patients’ preference for sports medicine physician characteristics, based on patient race and socioeconomic status.”
10. Consistency in your format for references. Some are not complete: see #23, 28 for example. Some publication dates are listed after the authors (see #8, #10) and most after the name of journal (as in #1 and #2)
This article presents a systematic review of studies addressing the concept of therapeutic alliance in the pediatric sports and orthopedics population. The article appears to be unique in this regard. It contributes to the literature by identifying factors associated with the development of therapeutic alliance. Eleven (11) articles were included from 2196 identified after search, sufficient for this review. The article follows PRISMA guidelines. The information presented is relevant and important in health care, especially with current emphasis on patient centered care and shared decision-making. The concept of therapeutic alliance for the physical therapist is essential, as the therapist functions as a coach and guide to wellness. Future directions sections reveals there is a wide range of investigations open regarding this important concept.
General concepts: One area of weakness is the bias identified in the articles included, with the use of Downs and Black checklist for reporting risk of bias assessment. All but one article score in the “poor” range with one article scoring “fair”. This speaks to the quality of existing data, however, not to this review. The authors should include a brief explanation of the bias found.
The review topic is important, timely and relevant to health care. The article is well-structured. I would recommend eliminate the ‘word cloud’ and combine some tables, but otherwise the tables and figures are relevant.
A gap in knowledge is identified, conclusions are supported. References are appropriate. Any recommendations to the authors should be considered minor and are easily addressed by them.
Reviewer 3 Report
The paper is a systematic review on Factors that Foster Therapeutic Alliance in Pediatric Sports and Orthopedics.
I found a good and precise methodology in the choice of the manuscripts, tables help readers to understand content and the several considerations.
The discussion of the literature is good, only I have a suggestion to add some considerations on the fact that also age could be a factor to be studied as impacting on therapeutic alliance. The age could be also a variable to study both of the adolescent and the health professional.
Conclusions are also really short, could be possible to add some more considerations for the health professionals?
The paper is clear and well written.
